# Augmenting Large Language Models with Knowledge Graphs for Domain-Specific Applications

Sushanta Mohapatra[1] and Tosin Adewumi[1]

[1]Luleå University of Technology
{sushanta.mohapatra@student.ltu.se, tosin.adewumi@ltu.se}

## 1 Abstract

Large language models (LLMs) models demonstrate impressive capabilities in generating human-like text and handling general-purpose queries. However, their application in specialized domains, such as supply chain management (SCM), remains challenging due to limitations in understanding domain-specific terminology and concepts. This research explores the integration of Knowledge Graphs (KGs) into Retrieval Augmented Generation (RAG) pipelines to enhance the performance of LLMs in domain-specific tasks. **We introduce a novel benchmark dataset for SCM**, covering eight supply chain functions and thirteen distinct categories of questions. The results of this study demonstrated that the KG integration improved performance compared to traditional RAG approaches, with smaller models achieving notable gains that reduced the performance gap with larger models.

## 1 Introduction

Large Language Models (LLMs) can answer questions and generate human-like text. However, they face significant challenges in specialized domains such as supply chain management (SCM), which involves specific terminology and complex processes unique to various organizations. Modern SCM operates within a dynamic global environment that requires effective coordination among multiple stakeholders. While LLMs have the potential for reasoning and problem-solving, their static general knowledge limits their effectiveness in addressing the intricacies of SCM. [1–4] To enhance LLMs, retrieval-augmented generation (RAG) frameworks have been proposed that integrate external knowledge to improve response accuracy. However, traditional RAG approaches often rely on basic vector similarity, which can result in incomplete or inconsistent information retrieval. By grounding LLMs in factual knowledge, KGs can improve the accuracy and relevance of generated content. [5–7]

This research work investigates knowledge augmentation of LLMs with KGs for domain-specific applications. It attempts to address limitations related to complex reasoning and domain-specific concepts in order to improve real world applications of LLMs. The **research question** of this study is 'How can the accuracy and context-awareness of LLMs be improved with the integration of KGs for decision-making processes and real-world applications in SCM? Hence, the goals of this project are:

- Investigate various strategies for integrating KGs into the RAG pipeline to enhance its functionality and effectiveness.

- Develop a framework to enrich LLMs with KGs, enabling them to better manage and understand the specific contexts and terminologies relevant to SCM.

## 2 Methods

The methodology used in this research work includes data acquisition and preparation, KG construction, solution development, evaluation, and critical analysis of results.

### 2.1 Data Acquisition and Preparation

This study utilizes two primary datasets: a novel supply chain benchmark dataset and the open LTU Chatbot QA dataset [8]. A novel supply chain benchmark dataset was developed to capture real-world SCM challenges. The questions in the dataset were divided into two parts: generic questions and organization-specific questions. Each part was further organized by eight SCM functions and 13 question categories. In total, 208 questions were curated for this dataset [2 (groups) x 8 (departments) x 13 (question categories)]. The LTU Chatbot QA dataset, originally by Werkman [8], was utilized after minor modifications. To better evaluate the retrieval capabilities of RAG systems, changes were done to decouple the direct association between questions and their corresponding knowledge source texts.

### 2.2 Knowledge Graph Construction

Three KGs were created based on distinct knowledge sources: 1) facts from the LTU website, 2)

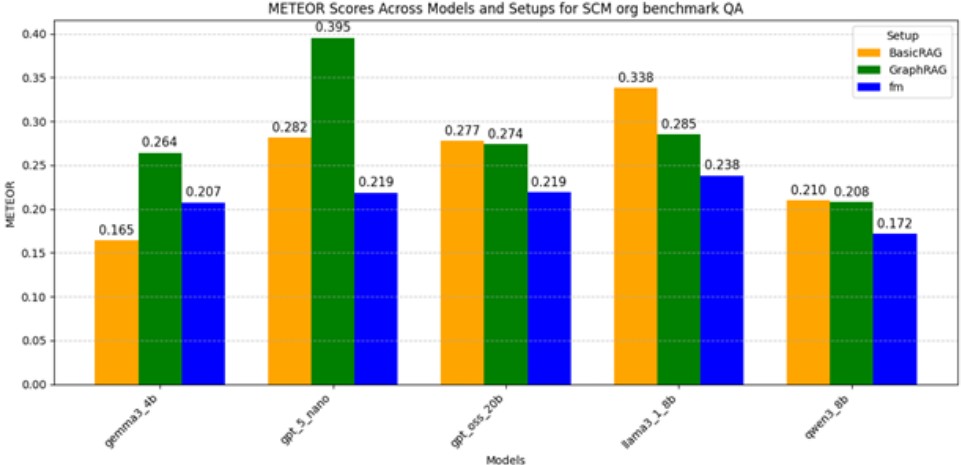

**Figure 1.** Performance across models for SCM Organization specific QA dataset (METEOR score)

generic SCM knowledge from publicly available literature, and 3) synthetically created internal process documentation for the fictional organization FoSCwAI AB. The KGs were constructed using GPT-4o-mini and involved several steps, including retrieving triplets from source texts, mapping these extracted triplets into a base ontology in JSON format, and iteratively refining the structure.

## 2.3 Solution Development and Evaluation

Five state-of-the-art LLMs were selected for experimentation: gemma3 (4b), qwen3 (8b), llama3.1 (8b), gpt-oss (20b), and GPT-5 (nano). Three solution pipelines were developed: a foundation model pipeline for direct question answering, a standard RAG pipeline utilizing vector similarity, and a KG-integrated RAG pipeline that enhances retrieval by incorporating KG entities. The Supply Chain Knowledge Augmentation and Enrichment (SC-KAE framework) was developed to improve knowledge retrieval and reasoning for complex SCM related queries. Evaluation was done using metrics such as ROUGE and METEOR, as well as truthfulness scores assessed by LLM-based evaluation.

## 3 Results and Discussion

The KG-integrated RAG approach outperformed other approaches in organization-specific contexts, improving both answer quality and alignment with knowledge bases. Figure 1 is a bar chart of performance on the SCM dataset. Similar observations were made with the LTU Chatbot QA dataset. This indicates that KGs can improve a model's ability to ground its outputs in structured, domain-relevant knowledge. While VectorRAG relies on unstructured text, KG-integrated RAG provides richer context,

resulting in better performance.

However, in broader open world contexts like the generic SCM QA benchmark, its advantages are less consistent, often trailing behind foundation models. Smaller, lightweight models benefited more from KG integration, showing marked improvements in truthfulness and performance. This suggests that with a robust KG, lightweight models can compete with larger ones, making deployments more cost-effective. Limitations include dependency on KG completeness and increased latency.

## 4 Conclusion

The study investigated the integration of KGs within RAG pipelines for domain-specific QA. The proposed KG integrated RAG framework, combining semantic entity linking, subgraph extraction, and LLM-based reasoning, demonstrably improves answer relevance, lexical overlap, and truthfulness compared to standard vector-based retrieval approaches. Our findings affirm the significant potential of KG integration to enhance grounding and factuality. This work lays a good foundation for future research in ontology-driven, retrieval-augmented AI systems in domain specific context, with promising applications in SCM, academic and other domains. Although the results of the study are promising, challenges remain, as the accuracy of the system strongly depends on the completeness and quality of the KG, prompting future efforts to optimize KG construction and improve semantic entity linking.

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
