# OpenReview forum: "Augmenting Large Language Models with Knowledge Graphs for Domain-Specific Applications"
_NLDL.org/2026/Abstracts_Track — NLDL 2026 Abstracts_

### Official Review · Reviewer_VSfa · 2025-10-28

**Soundness:** 3
**Correctness:** 2
**Rating:** 2
**Confidence:** 3

**Summary:**

The study applies Knowledge-Graph enhanced RAG for domain specific QA in Supply-Chain-Management. It introduces a benchmark for this task based on an adapted dataset and compares the method versus a model baseline, traditional (vector only) RAG, and the KG extended RAG. The reported results indicate better performance on the METEOR score on 2/5 models, but poorer performance on 3/5 models.

**Strengths:**

- Interesting and relevant application of KG enhanced RAG in a particular domain
- Improved performance over naive RAG in some cases (on the METEOR dataset)
- Introduces a new benchmark dataset for RAG on SCM-data
- Multi source KG construction and acknowledges limitations in KG completeness

**Weaknesses:**

- ROGUE/METEOR are perhaps not the best metrics to assess performance in this context
- Only results on METEOR are reported and claimed superior performance is not consistent across models.
- An example of benchmark instances would be interesting to understand the test cases
- Potential data leakage between the synthetic documentation / LLM generated KG and the eval benchmark set
- No details on retrieval settings are specified

---

### Official Review · Reviewer_hqqv · 2025-11-01

**Soundness:** 3
**Correctness:** 2
**Rating:** 4
**Confidence:** 4

**Summary:**

The abstract aims to improve the performance of LLMs by utilizing knowledge graphs (KGs) into retrieval augmented generation (RAG) models in the specialized domain of supply chain management (SCM). It also introduces a novel dataset for this purpose, and trains and evaluates models both on this dataset and an existing one. The KG was created both with real information from the researchers university’s website and synthetic data, and was processed using an LLM. The models using both LLMs and KGs were then trained and evaluated using five different LLMs. The results showed that the models utilizing the KGs generally performed better than the LLMs without them, but that these results were weaker in a broader SCM context.

**Strengths:**

- The abstracts introduce ways to incorporate knowledge into LLMs by using knowledge graphs, which can help make LLMs more truthful, which is a known and important issue in machine learning.
- A novel dataset is introduced, which makes it easier for other researchers to benchmark their methods.
- Five different LLMs are utilized on two different models, providing a rich set of empirical results.
- According to the empirical data, the models utilizing the KGs perform better than the ones without.
- The abstract also addresses some issues, like poorer performance in a broader context and some limitations.

**Weaknesses:**

- Figure 1 needs substantial improvement.
  - First of all, the resolution is way too low, making it difficult to simply read the content. The figure should ideally be included in a PDF format, or at least a much higher resolution PNG format.
  - It is not clear from the figure’s description what the different models are. I assume that the "BasicRag" does not utilize the KGs while the "GraphRag" does, but this should be explicitly stated. Furthermore, it is not clear at all what the third model is or even how it reads. Is this "fm", "lm", what is it? Please state information like this clearly in the figure’s description.
  - The figure only contains a single metric on a single dataset, which is only a small subset of the evaluation. It is reasonable to include only a part of the results in the abstract version of the research, but it should be possible to make a more information-dense figure, including more metrics or both datasets. Given only this figure, it is not possible to know if the results have been cherry-picked.
- Since the novel dataset is a main part of the contributions, the abstract could benefit from more details about this dataset. Specifically, how were the questions created and chosen? Also, it is mentioned that ChatGPT 4o-mini was used in the construction of the KG. Why was this done? As the KG should be the source of truth, why is this not constructed in a logical or manual way that ensures that the KG remains correct? LLMs may introduce mistakes in the KGs, which would make the whole dataset less useful.
- Since models such as Chat-GPT 4o and GPT-5 are used, which are continuously changed and updated, the authors should state which date or specific version that was utilized for reproducibility.
- The authors state that the advantages of the new models are less evident in a broader context, but it is not clear from which empirical results this claim is based on.
- The specific architecture of the RAG models utilizing KGs could be explained in more detail, but this is acceptable given the format of the research is in an abstract.
- Line 021 should 029 should be backed up by sources. Since the setting for the whole problem is introduced, the authors should be able to refer to existing material that covers these topics. These references should ideally have importance in the machine learning community, which would then imply that the work in this abstract is important.
- The research question in lines 047 to 050 is not properly quoted. It contains a single quote in the beginning and no ending quote.
- Lines 096 to 097 should have references to the models.

---

### Official Review · Reviewer_Taw3 · 2025-11-03

**Soundness:** 3
**Correctness:** 3
**Rating:** 4
**Confidence:** 3

**Summary:**

This paper looks at how knowledge graphs can be used within retrieval augmented generation (RAG) pipelines to make large language models more reliable for domain specific use, focusing on supply chain management (SCM). The authors also introduce a new SCM dataset and compare several model setups, baseline LLMs, standard RAG, and RAG enhanced with a KG component. They find that KG integration helps smaller models produce more grounded and relevant responses, reducing the performance gap with larger models.

**Strengths:**

- The problem is timely, trying to make LLMs more useful in vertical domains, and this can be a nice contribution to that area.
- The experimental setup is clear and structured.
- I like the practical focus on smaller models and cost efficiency.
- The results, though preliminary, are encouraging.

**Weaknesses:**

- The evaluation section is light on quantitative details and a few more metrics or examples would strengthen the claims.
- It’s not entirely clear if or when the SCM dataset will be released, which limits reproducibility.
- The paper could spend more time analyzing why KG integration helps.

---

### Decision · Program_Chairs · 2025-11-05

**Decision:**

Accept

**Comment:**

The reviewers found the abstract borderline, yet the PCs believe it will be of interest to the community and should have the opportunity be presented.